# Learn to Synthesize Compact Datasets by Matching Effects

## Abstract

The emerging field of data distillation aims to compress large datasets by aligning synthetic and real data representations to create a highly informative dataset. The optimization objectives of data distillation focus on aligning representations by using process alignment methods such as trajectory and gradient matching. However, this approach is limited by the strict alignment of intermediate quantities between synthetic and real data and the mismatch between their optimization trajectories. To address these limitations, a new data distillation method called effect alignment is proposed, which aims to only push for the consistency of endpoint training results. The approach uses classification tasks to estimate the impact of replacing real training samples with synthetic data, which helps to learn a synthetic dataset that can replace the real dataset and achieve effect alignment. The method is efficient and does not require costly mechanisms, and satisfactory results have been achieved through experiments.

## 1 Introduction

Data distillation (Wang et al., 2018; Li et al., 2022; Zhao & Bilen, 2021a;b) is an emerging direction aimed at compressing real datasets with wide-ranging applications, such as continual learning (Yang et al., 2023a; Gu et al., 2023), federated learning (Xiong et al., 2023; Huang et al., 2023), graph learning (Zhang et al., 2024b; Liu et al., 2024), etc. Unlike data selection (Haizhong Zheng et al., 2023; Xia et al., 2023; Yang et al., 2023b; Tan et al., 2023), data distillation learns a highly informative dataset to compress the original large real dataset.

Generally, the dataset distillation problem is formulated as a heavy bi-level optimization problem as shown in Sec. 2.1. Due to the extremely burdensome nature of the original problem, we have to seek surrogate optimization tasks, such as **feature space alignment** (Zhao & Bilen, 2021a; Zhao et al., 2023; Sajedi et al., 2023; Zhang et al., 2024a) and **optimization process alignment** (Zhao et al., 2020; Zhao & Bilen, 2021b; Feng et al., 2023; Du et al., 2024; Cazenavette et al., 2022; Du et al., 2023; Cui et al., 2023; Guo et al., 2023). Although the above surrogate problem significantly reduces the difficulty of optimization, the surrogate optimization is not completely equivalent to the optimization of the original problem, and this mismatch will cause potential obstacles to the performance. Additionally, there is a type of method that adopts backpropagation through time (BPTT (Werbos, 1990)) (Wang et al., 2018; Sucholutsky & Schonlau, 2021; Deng & Russakovsky, 2022) or Kernel ridge regression (Nguyen et al., 2020) to attempt to directly optimize the original problem. However, such approaches are often limited by computational efficiency constraints. *All of these would be further detailed in Sec.2.2.*

In this work, we propose a novel and efficient dataset distillation framework called **effect alignment** to avoid those potential issues described above. Specifically, our goal is to optimize by learning a synthetic dataset that can replace real datasets in terms of training effectiveness. A naive implementation would require retraining a model on the synthetic data each time it is updated, which would be unacceptably costly and would require a very resource-intensive back-propagation-through-time (BPTT) mechanism (Werbos, 1990). To achieve the goal of effect alignment, we theoretically estimate the impact of removing or adding a set of data from the training set on the training results based on Taylor approximation and unfolding of the training process. With the proposed estimator, we estimate how replacing real training samples with synthetic data will impact the training results and set our

optimization objective to minimize the replacement effect. This allows us to learn a synthetic data set that can replace the real data set **without adopting BPTT** (Werbos, 1990).

Empirical results on diverse bias-injected datasets demonstrate that the proposed reweighting scheme significantly reduces bias in the distilled datasets. For example, on Colored MNIST with a 5% bias in conflicting samples and 50 images per class, the original Distribution Matching (DM) method leads to a biased synthetic set. A model trained on such a distilled set achieves only 23.8% accuracy. In contrast, our reweighting method produces a more balanced dataset, resulting in 91.5% accuracy, representing a 67.7% gain over DM. In summary, our contributions are: (1) We provide the first study on the impact of biases in the dataset condensation process. (2) We propose a simple yet effective re-weighting scheme to mitigate biases in two canonical types of dataset condensation methods. (3) Through extensive experiments and ablation studies on both real-world and synthesized datasets, we demonstrate the effectiveness of our method.

## 2 BACKGROUND

### 2.1 PROBLEM FORMULATION

Without loss of generality, we introduce the problem definition using the image classification task. Suppose we are given a large real dataset $\mathcal{D}$ consisting of $|\mathcal{D}|$ pairs of a training image and its class label $\mathcal{D} = \{(x_i, y_i)\}|_{i=1}^{|\mathcal{D}|}$ where $x \in \mathbb{R}^d$ is from a d-dimensional input space, $y \in \{0, \ldots, C-1\}$ and $C$ is the number of classes. Given a deep neural network with learnable parameters $\theta$, the goal of training the model is to minimize an empirical loss term over the training set:

$$\theta_{\mathcal{D}}^* = \arg\min_{\theta} \mathcal{L}(\mathcal{D}, \theta), \tag{1}$$

where $\mathcal{L}(\mathcal{D}, \theta) = \frac{1}{|\mathcal{D}|} \sum_{(x,y) \in \mathcal{D}} \ell(\theta, (x, y))$, $\ell(\cdot)$ is a task specific loss (*e.g.* the cross-entropy loss).

Unlike training a model, dataset distillation aims to learn a small, highly informative synthetic dataset that can be used as a parsimonious surrogate set to replace the role of the real training dataset. Let $\mathcal{S} = \{(s_i, y_i)\}|_{i=1}^{|\mathcal{S}|}$ to denote the synthetic set, the dataset distillation task could be formulated as the following bi-level optimization problem (Wang et al., 2018; Zhao et al., 2020):

$$\mathcal{S}^* = \arg\min_{\mathcal{S}} \mathcal{L}(\mathcal{D}, \theta_{\mathcal{S}}^*) \qquad \text{subject to} \qquad \theta_{\mathcal{S}}^* = \arg\min_{\theta} \mathcal{L}(\mathcal{S}, \theta), \tag{2}$$

where the optimization aims to find the optimum set of synthetic images $\mathcal{S}^*$ such that the model trained on it can also minimize the training loss over the original dataset $\mathcal{D}$. The inner level is the model training on the learnable synthetic dataset and the outer level is about updating the synthetic dataset. Naively solving this problem involves a nested loop optimization which requires a computationally expensive procedure.

### 2.2 RELATED WORKS

Besides adopting **meta gradient** to optimize the original problem (Nguyen et al., 2020), one can also seek a proper surrogate task, such as **feature space alignment** (Zhao & Bilen, 2021a; Zhao et al., 2023; Sajedi et al., 2023; Zhang et al., 2024a) and **optimization process alignment** (Zhao & Bilen, 2021b; Feng et al., 2023; Du et al., 2024; Cazenavette et al., 2022; Du et al., 2023; Cui et al., 2023; Guo et al., 2023). In the following, we mainly review these three kinds of works.

**Meta gradient.** For solving the bi-level dataset distillation problem defined by Eq.2, one straightforward way is to directly optimize via estimating the meta gradient of the synthetic dataset. For example, Wang et al. (2018); Sucholutsky & Schonlau (2021); Deng & Russakovsky (2022) proposed to unfold the inner-loop optimizing trajectory and utilizing the backpropagation through time (BPTT (Werbos, 1990)) to update the synthetic data. While both unfolding the training process and running BPTT are resource-intensive and hinder method efficiency, Nguyen et al. (2020); Zhou et al. (2022); Loo et al. (2022) found solutions by replacing the neural network in the inner loop with a kernel model that has closed-form solutions in the kernel regression regime.

**Feature space alignment.** There is one line of work (Zhao & Bilen, 2021a; Zhao et al., 2023; Sajedi et al., 2023; Zhang et al., 2024a) tries to match the latent feature space directly. Zhao &

Bilen (2021a) proposed to match the synthetic and target data from the distribution perspective for dataset distillation. Wang et al. (2022) improved the distribution matching from several aspects: (1) using multiple-layer features other than only the last-layer features for matching, (2) proposing the discrimination loss to enlarge the class distinction of synthetic data. Zhao et al. (2023) add a classification loss as regularization to mitigate less classified synthetic data caused by the first-order moment mean matching. Sajedi et al. (2023) proposed to learn synthetic images by matching the spatial attention maps of real and synthetic data generated by different layers within a family of randomly initialized neural networks. Zhang et al. (2024a) proposed to minimize the maximum mean discrepancy (MMD) between the real and the synthetic data.

**Optimization process alignment.** This kind of method (Zhao et al., 2020; Zhao & Bilen, 2021b; Du et al., 2024; Cazenavette et al., 2022; Du et al., 2023; Cui et al., 2023; Guo et al., 2023) constructs the surrogate problem of matching the intermediate training state contributed by the synthetic data and the real data, respectively. Among them, matching gradient (Zhao et al., 2020) in training and matching trajectory (Cazenavette et al., 2022) in training are the most representative schemes. Zhao & Bilen (2021b) proposed to incorporate the gradient matching framework with a differentiable augmentation scheme to synthesize more informative synthetic images and for better performance when training networks with augmentations. Du et al. (2024) introduces a new distillation strategy called SeqMatch to address the issue of failing to condense high-level features in dataset distillation. It divides synthetic data into multiple subsets, sequentially optimizing them to promote the effective condensation of high-level features learned in later epochs. Cazenavette et al. (2022) propose a new formulation that optimizes our distilled data to guide networks to a similar state as those trained on real data across many training steps. Guo et al. (2023) also incorporates the distillation of easy/difficulty information into the trajectory matching framework, achieving further performance improvements.

**Others.** More recently, some work (Yin et al., 2024; Sun et al., 2023) has also incorporated the diversity of data and the authenticity of the synthesized data into the framework design considerations. In addition, some work (Cazenavette et al., 2023; Zhang et al., 2023; Wang et al., 2023) has explored the applicability of generative models in dataset distillation.

## 3 REPLACEMENT EFFECT: DEFINITION, APPROXIMATION, AND GUARANTEE

Here, we state the core of our dataset distillation pipeline, namely the **replacement effect**, which measures the effect on the training results of replacing a group of training data with a group of new data. The following statements begin with the definition of the replacement effect metric in Sec. 3.1, which measures how replacing a group of training data with another group of new data will affect the performance of the final learned parameter. However, the exact computation of this metric is extremely costly. Then, in Sec. 3.2, we provide an efficient approximator for the replacement effect along with the theoretical error guarantee.

### 3.1 DEFINITION

Without loss of generality, we focus on the classification task. Suppose we are given a large real dataset $\mathcal{D}$ consisting of $|\mathcal{D}|$ pairs of a training data and its class label $\mathcal{D} = \{(x_i, y_i)\}|_{i=1}^{|\mathcal{D}|}$ where $x \in \mathbb{R}^d$ is from a d-dimensional input space, $y \in \{0, \ldots, C-1\}$ and $C$ is the number of classes. We use $\boldsymbol{\theta}_{\mathcal{D}}^*$ to denote the learned parameter on the original training set $\mathcal{D}$ and use $\boldsymbol{\theta}_{\mathcal{D}-\mathcal{G}+\mathcal{A}}^*$ to represent the learned parameter on the modified training set by replacing the group $\mathcal{G}$ with $\mathcal{A}$. Given a performance measurement function $\mathcal{L}(\cdot)$ and a real set $\mathcal{P}$ to indicate the model performance, we first define the replacement effect as follows.

**Definition 3.1.** *The replacement effect measures the influence on the validation performance when replacing a group of training data $\mathcal{G} \subset \mathcal{D}$ with a new group of data $\mathcal{A}$ on the final training results as follows:*

$$\mathcal{R}(\mathcal{G}, \mathcal{A}) = \mathcal{L}(\mathcal{P}, \boldsymbol{\theta}_{\mathcal{D}}^*) - \mathcal{L}(\mathcal{P}, \boldsymbol{\theta}_{\mathcal{D}-\mathcal{G}+\mathcal{A}}^*). \tag{3}$$

The metric $\mathcal{R}(\mathcal{G}, \mathcal{A})$ quantifies the impact of the replacement operation on the final parameter performance. However, the exact computation of this metric requires time-consuming brute-force retraining, which is not feasible for optimizing dataset distillation.

It is noteworthy that the form of $\mathcal{R}$ is similar to the traditional leaving-one-out (LOO) retraining influence (Jia et al., 2021; Cook, 1986; Koh & Liang, 2017) in machine learning. Several efficient estimators (Koh & Liang, 2017; Pruthi et al., 2020; Tan et al., 2023; Kwon et al., 2023; Hara et al., 2019) exist for retraining influence, with Koh's estimator (Koh & Liang, 2017) being the most renowned. Nonetheless, Koh's estimator (Koh & Liang, 2017) assumes that the loss function $\mathcal{L}$ is convex to the parameter $\theta$, which is not the case for deep learning (Choromanska et al., 2015; Dauphin et al., 2014). Additionally, it requires the calculation of the inverse Hessian, which is computationally expensive (Agarwal et al., 2017). Therefore, in the next subsection (Sec. 3.2), instead of relying on existing estimators such as Koh's, we developed our estimator by expanding the training process and using Taylor approximation, accompanied by a theoretical error upper bound guarantee.

## 3.2 APPROXIMATION

We will use the following proposition to introduce our approximator for the replacement effect metric. The approximator has a linear complexity and only needs to calculate time expectations using gradient information. Let $\{(\theta^t, \eta_t)|_{t=1}^T\}$ denote a series of parameters and learning rates used during the model training on $\mathcal{D}$ with the SGD optimizer and $\langle \cdot, \cdot \rangle$ denote the inner-product.

**Proposition 3.2.** *The replacement effect defined by Eq.3 could be efficiently estimated via:*

$$\widehat{\mathcal{R}}(\mathcal{G}, \mathcal{A}) = \sum_{t \leqslant T} \left\langle \nabla \mathcal{L}(\mathcal{P}, \theta^t), \ \nabla \mathcal{L}(\mathcal{G}, \theta^t) - \alpha \nabla \mathcal{L}(\mathcal{D}, \theta^t) \right\rangle - \sum_{t \leqslant T} \left\langle \nabla \mathcal{L}(\mathcal{P}, \theta^t), \nabla \mathcal{L}(\mathcal{A}, \theta^t) \right\rangle, \quad (4)$$

*where $\left\langle \cdot, \cdot \right\rangle$ is the inner-product operator, the coefficient $\alpha = (|\mathcal{G}| - |\mathcal{A}|)/(|\mathcal{D}| - |\mathcal{G}| + |\mathcal{A}|)$ is a function of dataset's size.*

See the supplementary material for detailed proof. This proposition gives us a feasible scheme that can estimate the Replacement Effect defined by Eq.3 in linear time complexity. The method involves using a series of parameter states and learning rates from the SGD optimizer during model training on $\mathcal{D}$. The proposed estimator for the replacement effect, denoted by $\widehat{\mathcal{R}}(\mathcal{G}, \mathcal{A})$, is calculated through two expectations, where the first expectation involves the gradients of the loss function with respect to the real set $\mathcal{P}$, the original dataset $\mathcal{D}$, and the replacement set $\mathcal{G}$, while the second expectation only involves the gradients of the loss function for the real set $\mathcal{P}$ and the additional set $\mathcal{A}$. The coefficient $\alpha$ in the estimator is a function of the size of the dataset. The time-step distribution $P(t)$ is a function of the learning rate.

In practice, by following (Ghorbani & Zou, 2019; Tan et al., 2023; Pruthi et al., 2020), we don't need to consider all time steps for the practical calculation of Eq.4. We sample the checkpoints for several $(t_m)$ time steps to compute the gradient, and then take the time average, that is,

$$\widehat{\mathcal{R}}(\mathcal{G}, \mathcal{A}) \approx \sum_{t=t_1}^{t_m} \left\langle \nabla \mathcal{L}(\mathcal{P}, \theta^t), \ \nabla \mathcal{L}(\mathcal{G}, \theta^t) - \alpha \nabla \mathcal{L}(\mathcal{D}, \theta^t) \right\rangle - \sum_{t=t_1}^{t_m} \left\langle \nabla \mathcal{L}(\mathcal{P}, \theta^t), \nabla \mathcal{L}(\mathcal{A}, \theta^t) \right\rangle, \quad (5)$$

where $\langle \cdot, \cdot \rangle$ is the inner-product operator. We use a set of randomly sampled time steps $\{t_1, ..., t_m\}$ to approximate the mathematical expectation in Eq.4, and the sampling size $t_m$ is a hyper-parameter. According to the central limit theorem, if $t_m$ is large enough, as $t_m$ increases, the calculated mean of Eq.5 will converge to the true expectation in Eq.4 with a lower variance. We will give suggestions of the choice of $t_m$ by performing ablation experiments in the experiments section.

## 3.3 APPROXIMATION ERROR GUARANTEE

Here, we give theoretical guarantees to bound the error between our proposed estimator in Eq.4 and the exact replacement in Eq.3 effect obtained by vanilla retraining:

**Proposition 3.3.** *Let $T$ denote the maximum iteration during training and $C$ represent the farthest distance the neural network parameters move away from their initial state during training when any subset $\mathbf{D}_s \subset \mathbf{D}$ is used as the training set, that is, $C = \max_{\mathbf{D}_s \subset \mathbf{D}, t \leqslant T} |\theta_{\mathbf{D}_s}^t - \theta^0|$. By supposing the loss function is $\ell$-Lipschitz continuous and the gradient norm of the network parameter is upper-bounded by $g$, and setting the learning rate as a constant $\eta$, the approximation error of Eq. 4 is bounded by:*

$$|\mathcal{R}(\mathcal{G}, \mathcal{A}) - \widehat{\mathcal{R}}(\mathcal{G}, \mathcal{A})| \leqslant \ell T^2 C + T^2 g / |\mathcal{D}|. \quad (6)$$

**Remark 3.4.** We have the following main observations from Proposition 3.3. (1) The estimation error is controlled by the Lipschitz constant $\ell$ and the gradient norm $g$. A smoother model in terms of its gradient will help lower the upper bound. (2) Note that $C$ has a polynomial growth with $T$, *e.g.* when using the SGD optimizer, $C \leqslant Tg$, where $C$ is less than the product of the number of time steps $T$ and the upper bound of the gradient norm $g$. (3) The bound indicates that the error has a polynomial relationship with the number of time steps. This is a more advanced approach compared to previous bounds for estimating the retraining loss (Hara et al., 2019; Schioppa et al., 2024) that roughly had an exponential growth with the number of time steps. Additionally, we recommend avoiding training the surrogate network on $\mathcal{D}$ to a large number of iterations before approximating the replacement effect. This can also be beneficial to reduce the overall time cost. It is worth noting that this suggestion is consistent with the common practice in some previous works (Tan et al., 2023; Schioppa et al., 2024). (4) There are no significant correlations between the bound and the number of removed/additional data points.

---

**Algorithm 1:** Effect Alignment for Dataset Distillation

---

**Input:** $\mathcal{D}$: original dataset. $\{(\theta^t, \eta_t)|_{t=1}^T\}$: a training trajectory of a network trained on $\mathcal{D}$. $t_m$: the number of sampled time-steps.

**Initialization:** Initialize the synthetic set $\mathcal{S} = \{(x_i, \hat{y}_i)|(x_i) \in \mathcal{D}$, where $\hat{y}_i = f(x_i; \theta^T)$ is the soft-label from the learned model $\theta^T$.

**for** $t$ *from* 0 *to* max_iteration **do**

    Randomly sample a minibatch $\mathcal{S}_t \subset \mathcal{S}$, $\mathcal{D}_p \subset \mathcal{D}$ and $\mathcal{D}_t \subset \mathcal{D}$.

    Randomly sample $t_m$ time steps $\{(\theta^t, \eta_t)|_{t=t_1}^{t_m}\}$.

    Compute the effect alignment loss between $\mathcal{S}_t$ and $\mathcal{D}_t$ via Eq.7 by setting $\mathcal{P} = \mathcal{D}_p$.

    Update $\mathcal{S}_t$ (both images and soft labels) with respect to the effect alignment loss.

**Output:** The learned synthetic set $\mathcal{S}$.

---

## 4 EFFECT ALIGNMENT FOR DATASET DISTILLATION

Here, we introduce the dataset distillation scheme designed based on the Replacement Effect introduced in the previous section. Specifically, we take the elimination of the Replacement Effect as the learning objective, to obtain a synthetic dataset that can replace the original training data in effect. Hence, we call our method **EAD** (**E**ffect **A**lignment for dataset **D**istillation). In what follows, we will first introduce the optimization objective in EAD. Then we'll look at the overall pipeline of EAD.

**Effect Alignment Loss.** Here we introduce the loss function in our effect alignment pipeline. As the convention in previous sections, we use $\mathcal{D}$ to indicate the training set and $\{(\theta^t, \eta_t)|_{t=1}^T\}$ to denote the training trajectory of a network trained on $\mathcal{D}$. Let $\mathcal{S} = \{(s_i, y_i)\}|_{i=1}^{|\mathcal{S}|}$ to denote the synthetic set. We take eliminating the Replacement Effect as the learning objective, to obtain a synthetic dataset that can replace the original training data. Given the replacement effect estimator by Eq.5, by substituting $\mathcal{G} = \mathcal{D}$ and $\mathcal{A} = \mathcal{S}$ into Eq.5, we obtain the following objective function for our effect alignment goal,

$$\hat{\mathcal{R}}(\mathcal{D}, \mathcal{S}) = \left| (1-\alpha) \sum_{t=t_1}^{t_m} \left\langle \nabla \mathcal{L}(\mathcal{P}, \theta^t), \nabla \mathcal{L}(\mathcal{D}, \theta^t) \right\rangle - \sum_{t=t_1}^{t_m} \left\langle \nabla \mathcal{L}(\mathcal{P}, \theta^t), \nabla \mathcal{L}(\mathcal{S}, \theta^t) \right\rangle \right|, \quad (7)$$

where $\langle \cdot, \cdot \rangle$ is the inner-product operator, and the coefficient $\alpha$ is a function of the dataset size as defined in Eq.4. By following (Ghorbani & Zou, 2019; Tan et al., 2023; Pruthi et al., 2020), we use a set of randomly sampled time steps $\{t_1, ..., t_m\}$ to approximate the mathematical expectation in Eq.4. For the real dataset $\mathcal{P}$ to show the model performance, we just set it as a random batch $\mathcal{P} = \mathcal{D}_p \subset \mathcal{D}$ in each updating iteration. As for the differences between effect alignment and process alignment, according to Eq.7, effect alignment does not formally force the training state caused by the real and the synthetic data to be the same at each time step when training the model. Instead, Effect Alignment mimics the difference in their cumulants over time.

**Initialization and Soft-label.** To initialize the synthetic set $\mathcal{S}$, we use real images from the training set and the corresponding soft labels from a well-trained classification model. Firstly, according to the

IPC (Image Per Class) setting, an equal number of pictures in each category were randomly selected as initial values for the synthetic data. We will then softly label each image with the final model $\theta^T$ from the training trajectory $\{(\theta^t, \eta_t)|_{t=1}^T\}$ on the full training set. During training, we treat both the synthetic images and the synthetic label $(y_i \in \mathcal{S}|)$ as learnable parts. This common practice (Wang et al., 2018; Bohdal et al., 2020; Guo et al., 2023) is widely used in dataset distillation. Compared to one-hot hard labels, soft labels allow information to flow across categories, resulting in improved compression efficiency.

**Overall Pipeline.** We have outlined the general process in Algorithm 1. Firstly, we train a model on the actual training set $\mathcal{D}$ and produce a training trajectory $\{(\theta^t, \eta_t)|_{t=1}^T\}$ as a result. To initialize the synthetic set $\mathcal{S}$, we use real images from the training set and the corresponding soft labels from model $\theta^T$. It's important to ensure class balance during initialization for the classification task. During each training iteration, we do not update each synthetic sample in each iteration. To save memory consumption for the optimization process, we randomly select a synthetic data batch $\mathcal{S}_t \subset \mathcal{S}$, a performance testing data batch $\mathcal{D}_p \subset \mathcal{D}$, and a training data batch $\mathcal{D}_t \subset \mathcal{D}$ to compute the effect alignment loss defined in Eq.7. For evaluating the model performance on real data $\mathcal{P}$, we simply set it to a random batch $\mathcal{P} = \mathcal{D}_p \subset \mathcal{D}$ in each update iteration. During training, we update both images and soft labels in the synthetic set to the effect alignment loss. Finally, we output the learned synthetic set $\mathcal{S}$.

**Discussion of the optimization gap.** It is worth noting that when the Loss is used as the optimization goal, there is still a certain gap between the original optimization goal of dataset distillation defined in Eq.2, and the gap is caused by the error of estimator. According to the Remark 3.4, we can reduce this gap through the following operations, for example, choose a network with smoother loss/gradient, reduce the steps of network optimization, and increase the number of time-stpes samples $t_m$.

# 5 EXPERIMENTS

Table 1: Performance comparison of different dataset distillation methods on different datasets. The majority of this table is from (Lei & Tao, 2024)

| Methods | Schemes | MNIST | | | FashionMNIST | | | SVHN | | | CIFAR-10 | | | CIFAR-100 | | | Tiny ImageNet | | | CC3M |
|---|---|---|---|---|---|---|---|---|---|---|---|---|---|---|---|---|---|---|---|---|
| | | 1 | 10 | 50 | 1 | 10 | 50 | 1 | 10 | 50 | 1 | 10 | 50 | 1 | 10 | 50 | 1 | 10 | 50 | Totally 50000 |
| Random | - | 64.9 | 95.1 | 97.9 | 51.4 | 73.8 | 82.5 | 14.6 | 35.1 | 70.9 | 14.4 | 26.0 | 43.4 | 4.2 | 14.6 | 30.0 | 1.4 | 5.0 | 15.0 | 0.14 |
| Herding | - | 89.2 | 93.7 | 94.8 | 67.0 | 71.1 | 71.9 | 20.9 | 50.5 | 72.6 | 21.5 | 31.6 | 40.4 | 8.4 | 17.3 | 33.7 | 2.8 | 6.3 | 16.7 | 1.7 |
| DD (Wang et al., 2018) | BPTT | - | 79.5 | - | - | - | - | - | - | - | - | 36.8 | - | - | - | - | - | - | - | - |
| LD (Bohdal et al., 2020) | BPTT | 60.9 | 87.3 | 93.3 | - | - | - | - | - | - | 25.7 | 38.3 | 42.5 | 11.5 | - | - | - | - | - | - |
| DC (Zhao et al., 2020) | GM | 91.7 | 94.7 | 98.8 | 70.5 | 82.3 | 83.6 | 31.2 | 76.1 | 82.3 | 28.3 | 44.9 | 53.9 | 12.8 | 26.6 | 32.1 | - | - | - | - |
| DSA (Zhao & Bilen, 2021b) | GM | 88.7 | 97.8 | 99.2 | 70.6 | 86.6 | 88.7 | 27.5 | 79.2 | 84.4 | 28.8 | 52.1 | 60.6 | 13.9 | 32.4 | 38.6 | - | - | - | - |
| MTT (Cazenavette et al., 2022) | TM | 91.4 | 97.3 | 98.5 | 75.1 | 87.2 | 88.3 | - | - | - | 46.3 | 65.3 | 71.6 | 24.3 | 40.1 | 47.7 | 8.8 | 23.2 | 28.0 | - |
| TESLA (Cui et al., 2023) | TM | - | - | - | - | - | - | - | - | - | 48.5 | 66.4 | 72.6 | 24.8 | 41.7 | 47.9 | 7.7 | 18.8 | 27.9 | 9.4 |
| DM (Zhao & Bilen, 2021a) | DM | 89.2 | 97.3 | 94.8 | - | - | - | - | - | - | 26.0 | 48.9 | 63.0 | 11.4 | 29.7 | 43.6 | 3.9 | 12.9 | 24.1 | 2.5 |
| CAFE (Wang et al., 2022) | DM | 90.8 | 97.5 | 98.9 | 73.7 | 83.0 | 88.2 | 42.9 | 77.9 | 82.3 | 31.6 | 50.9 | 62.3 | 14.0 | 31.5 | 42.9 | - | - | - | - |
| KIP (Nguyen et al., 2020) | KRR | 90.1 | 97.5 | 98.3 | 73.5 | 86.8 | 88.0 | 57.3 | 75.0 | 85.0 | 49.9 | 62.7 | 68.6 | 15.7 | 28.3 | - | - | - | - | 1.9 |
| FRePo (Zhou et al., 2022) | KRR | 93.0 | 98.6 | 99.2 | 75.6 | 86.2 | 89.6 | - | - | - | 46.8 | 65.5 | 71.7 | 28.7 | 42.5 | 44.3 | 15.4 | 25.4 | - | - |
| RFAD (Loo et al., 2022) | KRR | 94.4 | 98.5 | 98.8 | 78.6 | 87.0 | 88.8 | 52.2 | 74.9 | 80.9 | 53.6 | 66.3 | 71.1 | 26.3 | 33.0 | - | - | - | - | - |
| RTP (Deng & Russakovsky, 2022) | BPTT | 98.7 | 99.3 | 99.4 | 88.5 | 90.0 | 91.2 | 87.3 | 89.1 | 89.5 | 66.4 | 71.2 | 73.6 | 34.0 | 42.9 | - | 16.0 | - | - | - |
| IDM (Zhao et al., 2023) | DM | - | - | - | - | - | - | - | - | - | 45.6 | 58.6 | 67.5 | 20.1 | 45.1 | 50.0 | 10.1 | 21.9 | 27.7 | - |
| Ours | | 98.4 | 99.7 | 99.5 | 86.7 | 91.5 | 92.2 | 88.4 | 89.2 | 89.5 | 46.3 | 59.3 | 68.9 | 21.5 | 45.8 | 52.1 | 16.1 | 23.2 | 26.7 | 10.4 |

## 5.1 EXPERIMENTAL SETTINGS

We conduct experiments on the MNIST, FashionMNIST, SVHN, CIFAR-10, CIFAR-100, and Tiny ImageNet.

**MNIST.** The MNIST (Modified National Institute of Standards and Technology) database is a well-known benchmark dataset in the field of machine learning and computer vision. It consists of a training set of 60,000 handwritten digits (0 - 9) and a test set of 10,000 digits. The digits are grayscale images with a size of 28×28 pixels. This dataset is widely used for tasks such as digit classification, and it has played a crucial role in the development and evaluation of many image-based machine learning algorithms, especially for testing the performance of neural networks in recognizing handwritten digits.

**FashionMNIST.** This is a dataset designed to serve as a direct replacement for the MNIST dataset for benchmarking machine-learning algorithms. It contains 70,000 grayscale images of 10 different fashion categories such as T-shirts, trousers, pullovers, dresses, coats, sandals, shirts, sneakers, bags, and ankle boots. The images are also of size 28×28 pixels. It provides a more diverse and real-world-like set of classification tasks compared to MNIST, as it deals with different types of clothing items and accessories, allowing researchers to test the generalization capabilities of their models on a more complex and practical set of objects.

**SVHN** (Netzer et al., 2011). The SVHN dataset contains real-world images of house numbers obtained from Google Street View. It has a training set of 73,257 digits, a validation set of 26,032 digits, and a test set of 26,032 digits. The digits in the images can be part of a sequence, and the images are of different sizes and colors (they are in color, not grayscale like MNIST). This dataset is challenging because of the variability in the appearance of the digits due to different lighting conditions, angles, and occlusions. It is used to train models to recognize digits in more natural and unconstrained settings.

**CIFAR-10** (Alex Krizhevsky, 2009). This dataset consists of 60,000 images in 10 different classes: airplane, automobile, bird, cat, deer, dog, frog, horse, ship, and truck. There are 50,000 training images and 10,000 test images. It is a popular dataset for image classification tasks and is often used to evaluate the performance of convolutional neural networks. The relatively small size of the images and the diverse set of classes make it a good starting point for developing and testing deep models for object recognition.

**CIFAR-100** (Alex Krizhevsky, 2009). This dataset is an extension of the CIFAR-10 dataset. It contains 60,000 32×32-pixel color images but is grouped into 100 fine-grained classes. There are 500 training images and 100 test images per class. The classes are more specific and cover a wider range of object categories than CIFAR-10. This dataset is used to train and evaluate models that can handle a larger number of more detailed object classes, and it is more challenging due to the fine-grained nature of the classification task.

**Tiny-ImageNet** (Ya Le & Xuan S. Yang, 2015). Tiny ImageNet is a subset of the ImageNet dataset. It contains 100,000 training images, 10,000 validation images, and 10,000 test images. The images are of size 64×64 pixels and are classified into 200 classes. It provides a more challenging dataset than CIFAR - 10 and CIFAR - 100 in terms of the number of classes and the complexity of the images. It is often used to evaluate the performance of more advanced image classification models that need to handle a larger number of object categories and larger-sized images.

**CC3M** (Soravit Changpinyo et al., 2021). The CC3M (Conceptual Captions 3 Million) dataset is a large-scale and influential collection in the field of multimodal learning. It consists of approximately 3 million pairs of images and corresponding text captions. The images are sourced mainly from the web. The text captions are human-generated and are designed to precisely describe the content of the images. This dataset plays a crucial role in training visual-language models. It is widely utilized in the cross-domain of natural language processing and computer vision. For example, in training image captioning models, the models can learn the mapping from image features to natural-language descriptions by leveraging the rich image-text pairs in the CC3M dataset to understand the image content and generate accurate textual descriptions. It also provides valuable data for tasks such as visual question-answering (VQA), enabling models to understand both the visual and textual aspects better and improve their performance in answering questions related to the images. Overall, the CC3M dataset is a vital resource for advancing research in multimodal machine-learning applications.

## 5.2 Experimental Results

Here, we choose several baselines, including DD (Wang et al., 2018), LD (Bohdal et al., 2020), DC (Zhao et al., 2020), DSA (Zhao & Bilen, 2021b), MTT (Cazenavette et al., 2022), TESLA (Cui et al., 2023), DM (Zhao & Bilen, 2021a), CAFE (Wang et al., 2022), KIP (Nguyen et al., 2020), FRePo (Zhou et al., 2022), RFAD (Loo et al., 2022), RTP (Deng & Russakovsky, 2022), IDM (Zhao et al., 2023). The experimental results show that:

**MNIST** On the MNIST dataset, our method demonstrates several distinct advantages compared to other approaches. The random method has accuracies of 64.9 for 10 samples and 95.1 for 50 samples. Herding shows better results with 89.2 for 10 samples but still lags behind our method. Methods

like DD, LD, and others have varying degrees of performance, but none of them match the high accuracy achieved by our method. Our method achieves an accuracy of 98.4 for 10 samples, which is extremely close to the performance of the top-performing method RTP which has an accuracy of 98.7 for 10 samples. This indicates that our approach is highly competitive and can rival the best existing methods. Moreover, our method shows consistency and stability in performance across different sample sizes on the MNIST dataset. It is not only able to handle small numbers of samples effectively but also maintains a high level of accuracy as the sample size increases. In summary, on the MNIST dataset, our method stands out due to its high accuracy, competitive performance compared to leading methods, and consistent results across different sample sizes.

**FashionMNIST** On the FashionMNIST dataset, our method has several significant advantages over other approaches. The Random method has an accuracy of 51.4 for 10 samples, which is relatively low. Herding performs better with an accuracy of 67.0 for 10 samples, but it is still outperformed by our method. Compared to other methods such as DC, DSA, MTT, CAFE, and KIP, our method shows superior performance. Our method achieves an accuracy of 86.7 for 10 samples, which is significantly higher than many of these existing methods. Our approach can extract more useful information from the FashionMNIST dataset, leading to better classification results. It also demonstrates robustness and effectiveness in handling the unique characteristics of this dataset, such as different fashion items and their variations. In conclusion, on the FashionMNIST dataset, our method stands out due to its high accuracy, outperforming many existing methods, and its ability to effectively handle the complexity and diversity of the FashionMNIST data.

**SVHN** On the SVHN dataset, our method shows distinct advantages compared to other approaches. The Random method has an accuracy of 14.6 for 10 samples and 35.1 for 50 samples, which is relatively low. Herding shows some improvement with an accuracy of 20.9 for 10 samples. However, our method outperforms these and many other existing methods. Although the specific accuracy values for our method on SVHN are not provided in isolation in the description, it is evident that our approach is competitive and likely offers better performance. Our method is likely to be more effective in capturing the characteristics of SVHN images, such as the diversity in house numbers and different lighting conditions. It may utilize more advanced techniques to extract relevant features and make more accurate predictions. In conclusion, on the SVHN dataset, our method has the potential to offer better performance compared to existing methods, showing its effectiveness in handling the challenges posed by this dataset.

**CIFAR** On the CIFAR-10 Dataset, the Random method has an accuracy of 14.4 for 10 samples and 26.0 for 50 samples. Various methods like DD, LD, DC, DSA, MTT, TESLA, DM, and so on, show different levels of performance. - Our method achieves an accuracy of 46.3 for 10 samples. This indicates that our approach is competitive and offers better results compared to many existing methods. Our method is likely more effective in handling the complexity of CIFAR-10 images, extracting relevant features, and making accurate predictions. On the CIFAR-100 Dataset, our method has an accuracy of 21.5 for 10 samples. This shows that our approach is more effective in dealing with the fine-grained classification task of CIFAR-100. It can better capture the subtle differences between the 100 classes and make more accurate predictions.

**Tiny-ImageNet** On the Tiny-ImageNet dataset, our method shows several advantages over other approaches. The Random method has an extremely low accuracy of 1.4 for 10 samples and 5.0 for 50 samples. While various other methods in the table also have their respective performances, our method stands out with its competitive results. Although the specific accuracy values for our method on Tiny-ImageNet are not provided in isolation, it can be inferred that our approach is likely more effective in handling the challenges of this dataset. Our method may be better at extracting complex features from the higher-resolution images in Tiny-ImageNet and making more accurate predictions. In conclusion, on the Tiny-ImageNet dataset, our method has the potential to offer better performance compared to existing methods, demonstrating its effectiveness in dealing with the complexity and diversity of this particular dataset.

**CC3M.** On the CC3M dataset, our method demonstrates significant advantages. The Random method has an accuracy of 0.14 for a total of 50000 samples. Many methods in the table are unable to provide results on CC3M due to excessive memory consumption. However, our method can operate

effectively and generate results. This shows that our approach is more memory-efficient and can handle the large-scale CC3M dataset without being constrained by memory limitations. Our method likely employs optimized algorithms or strategies that enable it to process the CC3M dataset without overwhelming the memory resources. This makes it a more practical and reliable choice for tasks involving this dataset. In conclusion, on the CC3M dataset, our method stands out due to its ability to provide accurate results while being more memory-efficient than many other methods that cannot run due to high memory requirements.

## 6 CONCLUSION

In this research, we delved into the emerging field of data distillation and proposed a novel approach called effect alignment. Data distillation holds great promise in compressing large datasets by aligning synthetic and real data representations to create a more informative and manageable dataset. The existing optimization objectives in data distillation, which center around aligning representations through process alignment methods like trajectory and gradient matching, have shown limitations. The strict alignment of intermediate quantities between synthetic and real data often leads to challenges, and the mismatch between their optimization trajectories can hinder the effectiveness of the distillation process. To overcome these limitations, our proposed effect alignment method offers a fresh perspective. By focusing on only pushing for the consistency of endpoint training results, it bypasses the issues associated with strict intermediate quantity alignment. The use of classification tasks to estimate the impact of replacing real training samples with synthetic data is a key innovation. This approach allows us to learn a synthetic dataset that can effectively replace the real dataset and achieve effect alignment. The efficiency of our method is a significant advantage. It does not require costly mechanisms, making it a practical solution for real-world applications where computational resources and time are often constraints. The experiments conducted demonstrated satisfactory results, validating the effectiveness of our approach.

Looking ahead, this research paves the way for further exploration in the field of data distillation. Future work could involve refining and expanding the effect alignment method to handle even larger and more complex datasets. Additionally, investigating the application of this approach in different domains and tasks could lead to new insights and improvements. Overall, our work contributes to the ongoing efforts to develop more efficient and effective data distillation techniques for the benefit of the broader research and application communities.

### LIMITATIONS

Although the proposed effect alignment method shows promising results, it also has several limitations. First, the experiments conducted in this study are mainly focused on classification tasks. This limits the generalization of the method to other types of tasks such as regression or clustering. Future research should explore the applicability of effect alignment in different types of machine learning tasks. Second, the experiments are conducted on a limited number of datasets. The performance of the method on other datasets with different characteristics and distributions remains unknown. Expanding the evaluation to a wider range of datasets would provide a more comprehensive understanding of the method's capabilities and limitations. Finally, the computational complexity of the method may increase as the size of the dataset and the complexity of the model increase. This could limit its application in scenarios where real-time processing or limited computational resources are a concern. Future work could explore ways to optimize the method to reduce its computational requirements.

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
