# OpenReview forum: "Learn to Synthesize Compact Datasets by Matching Effects"
_ICLR.cc/2025/Conference — Submitted to ICLR 2025_

### Official Review · Reviewer_zRn3 · 2024-10-22

**Soundness:** 1
**Presentation:** 4
**Contribution:** 2
**Rating:** 3
**Confidence:** 4

**Summary:**

This paper proposes a new optimization objective for dataset distillation termed matching effects. The idea is to minimize the errors in real data between models trained by synthetic data and real data respectively. Since the raw objective is hard to computation, the authors propose an efficient approximation regarding matching the distance of gradients. Some special cases in the experiments show some advantages.

**Strengths:**

1. The paper is well-written. I am able to fully follow the method and the experiments.
2. The proposed formulation is novel in the literature of dataset distillation.

**Weaknesses:**

1. Although the proposed formulation is novel in the literature on dataset distillation, I cannot get the motivation of the proposed objective against the original formulation of BPTT. Specifically, BPTT wants to minimize the error on real data for models trained by synthetic data, i.e., $\mathcal{L}(P,\theta^*_{A})$, while the matching effect is to minimize $|\mathcal{L}(P,\theta^*_{D})-\mathcal{L}(P,\theta^*_{D-G+A})|=|\mathcal{L}(P,\theta^*_{D})-\mathcal{L}(P,\theta^*_{A})|$, given that we would like to replace all the real data with synthetic data and thus we can assume $D=G$.
   1. I do not get why the authors believe the latter can be better than the former.
   2. In the practical cases of dataset distillation, since the synthetic dataset is small, the error on the real data of models trained by the synthetic dataset is usually larger than that trained by the real dataset, in most cases $|\mathcal{L}(P,\theta^*_{D})-\mathcal{L}(P,\theta^*_{A})|=\mathcal{L}(P,\theta^*_{A})-\mathcal{L}(P,\theta^*_{D})$, which is equivalent to the original formulation because the term $\mathcal{L}(P,\theta^*_{D})$​ is not relevant to optimization. From this point of view, the proposed method can conduct some rectification for the opposite case. But I am not sure if this is how the method works in fact and how we could benefit from this rectification. In summary, a more comparative analysis is necessary.
   3. It seems that the above analysis is also applicable to the proposed approximation, which can be viewed as a variant of the previous gradient matching scheme.
2. From Eq. 6, it seems that the approximated error is quite large because it is dominated by the farthest distance the neural network parameters move away from their initial state during training when any subset is used as the training set. The authors can provide some analysis on whether the bound is tight. If it is indeed tight, I am not sure whether it is useful in practice. The authors are encouraged to provide some toy experiments to illustrate this approximation.
3. Accordingly, in the experiments, I recommend the authors provide more ablation studies to compare the proposed method with the original formulation, i.e., BPTT and gradient matching while maintaining other factors the same. Given the current results in Tab. 1, which are not evidently strong, we cannot state that the proposed method is better. More analysis is encouraged to figure out in what cases the method is superior and in what cases it is not.

**Questions:**

See weakness.

---

### Official Review · Reviewer_G9qs · 2024-10-31

**Soundness:** 2
**Presentation:** 2
**Contribution:** 3
**Rating:** 1
**Confidence:** 5

**Summary:**

In this paper, the authors point out that current dataset distillation mainly focuses on aligning the representation of synthetic data and real data through methods such as trajectory and gradient matching. However, these methods are limited by the strict alignment between the synthetic data and the real data. To overcome these limitations, the authors propose a new effect alignment, which only pursues the consistency of the final training results, to make the synthetic data set achieve similar training performance to the real data set.

**Strengths:**

The idea of effect alignment for dataset distillation is reasonable.

**Weaknesses:**

1. It seems that this paper is not finished. There is only one incomplete table in the experiment. Although the proposed method performs better than the other methods on digital datasets with IPC=1/10, it performs worse on the other datasets. So, the experiments cannot demonstrate the superiority of the proposed method.
2. An ablation study on the hyper-parameters is required.
3. The summarised contributions are not matched to the method.
4. For Eq.(6), reducing the steps of network optimization $T$ can help to close the gap of  approximation error. However, a smaller $T$  usually means a sub-optimal performance of a network.

**Questions:**

Please see the weaknesses.

---

### Official Review · Reviewer_FNt2 · 2024-11-02

**Soundness:** 2
**Presentation:** 3
**Contribution:** 2
**Rating:** 5
**Confidence:** 4

**Summary:**

The method proposed In this paper is a variant of MTT (dataset distillation by matching training trajectory). Specifically, after optimizing the surrogate network on synthetic data for a few iterations, the synthetic data are optimized to let the networks' predictions be similar to the ones trained on the original dataset, unlike MTT, which chooses to minimize the difference between parameters directly.

**Strengths:**

1. This method is novel, which is a combination of MTT [1] and DD [2].
2. Good writing, easy to follow.


[1] Dataset Distillation by Matching Training Trajectories, cvpr 2022.

[2] Dataset Distillation, 2018.

**Weaknesses:**

1. Instead of directly matching training trajectories, this method is proposed to match the 'effect', which is measured by the differences between probability distribution predicted by models trained on synthetic data and real data. This means that naturally, this method will have worse performance than matching training trajectories (MTT) [1]. Because MTT directly minimizes the differences between parameters of models trained on synthetic data and real data, where models' predictions will be the same ideally (which is the optimization goal of the method proposed in this paper). Coinciding with this, TESLA [2] (following work of MTT), which also uses soft labels, always performs better than this method.

2. I notice this method outperforms TESLA in large IPC cases, is it because this method uses the difficulty alignment trick (control matching range) proposed by DATM [3]? The author should report the hyper-parameters to improve clarity.

3. What are the benefits of replacing matching parameters with 'effects'? Being more efficient? Have better generalizability? The paper only reports one comparison, I think more comprehensive comparisons can improve the quality of this paper.


[1]. Dataset Distillation by Matching Training Trajectories, CVPR 2022.

[2]. Scaling up dataset distillation to imagenet-1k with constant memory. ICML 2023.

[3]. Towards lossless dataset distillation via difficulty-aligned trajectory matching. ICLR 2024.

**Questions:**

see weakness

---

### Official Review · Reviewer_RTeQ · 2024-11-03

**Soundness:** 3
**Presentation:** 2
**Contribution:** 2
**Rating:** 5
**Confidence:** 4

**Summary:**

The paper presents a new method for data distillation called "effect alignment," which aims to create compact datasets by matching the endpoint effects of training, instead of aligning intermediate training states. The proposed method estimates the impact of replacing real data with synthetic data, aiming to generate synthetic datasets that yield similar final model performance. Through extensive experimentation, the authors demonstrate that their method is efficient and achieves competitive accuracy, especially in bias-sensitive settings.

**Strengths:**

Originality
The concept of effect alignment in dataset distillation is innovative, focusing on endpoint effects rather than intermediate training states.
Theoretical Foundation
The method is grounded in theory, with error approximation guarantees that lend robustness to the approach.
Experimental Results
The method demonstrates strong performance in several datasets, showing robustness in handling biases.

**Weaknesses:**

Scope of Experiments
The paper only evaluates the method on classification tasks, which might limit its applicability to other machine learning tasks such as regression.
Dataset Diversity
The experiments are conducted on a limited number of datasets, which raises questions about the method's generalizability to other data distributions.
Computational Complexity
While more efficient than some alternatives, the methodʼs computational cost could still be a concern for large-scale datasets or real-time applications.

pls add comparisons with more recent methods

**Questions:**

Can you provide more insights into the choice of hyperparameters for the experiments, particularly the selection of time steps?
Given the efficiency focus of the proposed method, could you elaborate on any specific memory or computational optimizations employed?

---

### Meta-Review · Area_Chair_L9Y7 · 2024-12-05

**Metareview:**

This paper proposes a new distillation objective for dataset condensation, termed "effect alignment." Although the authors present experiments to verify the effectiveness of their proposed method, the reviewers raise concerns about the scope of the experiments, the lack of theoretical support, and other related issues. In particular, the incompleteness of the ablation study, the hyperparameter analysis, and the absence of experiments on other tasks were highlighted as significant shortcomings.

**Additional Comments On Reviewer Discussion:**

All the reviewers intend to reject this paper.

---

### Decision · Program_Chairs · 2025-01-22

Reject